# New Orbit Determination Method for GEO Satellites Based on BeiDou Short-Message Communication Ranging

Xiaojie Li [1,2,3], Rui Guo [1,2,3,*], Jianbing Chen [4], Shuai Liu [1], Zhiqiao Chang [1], Jie Xin [1], Jinglei Guo [1] and Yijun Tian [1]

1   Beijing Satellite Navigation Center, Beijing 100094, China
2   Shanghai Key Laboratory for Space Positioning and Navigation, Shanghai 200030, China
3   Shanghai Astronomical Observatory, Shanghai 200030, China
4   China Top Communication Co., Ltd., Beijing 100088, China
*   Correspondence: shimbarsalon@163.com

**Abstract:** The radio determination service system (RDSS), a navigation and positioning system independently developed by China, features services such as short-message communication, position reporting, and international search and rescue. The L-band pseudo-range and phase data are the primary data sources in precise orbit determination (POD) for geostationary Earth orbit (GEO) satellite in the BeiDou system, especially in the orbit manoeuvre period. These data are the only data sources in the POD for GEOs. However, when the pseudo-range and phase data is abnormal due to unforeseen reasons, such as satellite hardware failure or monitoring receiver abnormalities, the data abnormality leads to orbit determination abnormalities or even failures for GEOs, then the service performance and availability of the RDSS system are greatly degraded. Therefore, a new POD method for GEOs based on BeiDou short-message communication ranging data has gained research attention to improve the service reliability of the BeiDou navigation satellite system (BDS)-3, realising the deep integration of communication and navigation services of the BDS. This problem has not been addressed so far. Therefore, in this study, a new POD method for GEO satellites is investigated using high-precision satellite laser ranging (SLR) data and RDSS data. The SLR data are used as the benchmark to calibrate the time delay value of RDSS equipment, and RDSS data are only used in the orbit determination process by fixing the corrected RDSS time delay value, and the satellite orbit parameters and dynamic parameters are solved. Experimental analysis is conducted using the measured SLR and RDSS data of the BDS, and the orbit accuracy in this paper is evaluated by the precise ephemeris of the Multi-GNSS pilot project (MGEX) and SLR data. The results show that the orbit accuracy in the orbital arc and the 2-h orbital prediction arc for GEOs are 6.01 m and 6.99 m, respectively, compared with the ephemeris of MGEX, and the short-arc orbit accuracy after 4 h of manoeuvring is 11.11 m. The orbit accuracy in the radial component by SLR data is 0.54 m. The required orbit accuracy for GEO satellites in the RDSS service of the BDS-3 is 15 m. The orbit accuracy achieved in this paper is superior to that of this technical index. This method expands the application field of the RDSS data and greatly enriches the POD method for GEOs. It can be adopted as a backup technology for the POD method for GEOs based on RNSS data, significantly improving the service reliability of the BeiDou RDSS service.

**Keywords:** BeiDou system (BDS); geostationary earth orbit (GEO) satellite; radio determination satellite service (RDSS); satellite laser ranging (SLR); multi-GNSS pilot project (MGEX)

## 1. Introduction

The BeiDou system (BDS) is a navigation and positioning system developed in China that provides two service modes, the radio navigation satellite service (RNSS) and the radio determination satellite service (RDSS) [1–3]. The benefits of RDSS as a featured part of BDS are to provide regional short message communication, global short message

communication, fast location reporting, and international search and rescue (SAR) services. RDSS users can not only know where they are but also tell others where they are. They can also transmit information to other RDSS users and send distress signals to rescue coordination centres (RCC) in distress. Therefore, the RDSS is an important component and a distinctive feature of the BDS. With more than ten years of development, the application scope of the BeiDou RDSS has been continuously expanded, including aviation, maritime, land transportation, and emergency rescue, et al. Therefore, the RDSS is an important component and a distinctive feature of the BDS, as GPS, GLONASS, and Galileo do not feature this component [4–7].

Scholars at home and abroad have conducted extensive research on BeiDou short-message communication ranging data, including the RDSS positioning algorithm, signal design, generalised RDSS design and extension, and combined navigation of the RDSS and strap-down inertial navigation system (SINS). A series of research results have been achieved, and the connotation of the satellite navigation system has been enriched [8]. However, the deep integrated application of BeiDou short-message communication and BeiDou navigation services has not been realised yet. The short-message communication ranging data have not been applied to the field of BeiDou satellite orbit determination, and the orbit accuracy for BeiDou satellites based on RDSS data has not been systematically analysed by the measured short-message communication ranging data.

GEO satellites, as an indispensable part of the BDS constellation composition, bear not only the responsibility of communication transmission but also the task of enhancing the geometric dilution of precision (GDOP) [9,10]. At present, BeiDou GEO satellite orbit determination mainly adopts L-band multi-frequency pseudorange, phase data, and ka-band inter-satellite link data of the RNSS system. The Overlap Orbit Differences (OOD) method is used to evaluate the accuracy of the 3D position of its post-precision products, which is a 2 m level for the BDS-2 GEO satellite based on the L-band multi-frequency pseudorange and phase data [11,12], and about 24 cm for the BDS-3 GEO satellite in the orbital arc based on the L-band data and Ka-band inter-satellite link data [2,13,14]. The laser ranging (SLR) data internationally jointly measured were used to evaluate the orbit accuracy in radial direction, which was 40 cm for the BDS GEO satellites [15–17].

In the case of abnormal RNSS and inter-satellite link data due to uncontrollable factors such as satellite navigation unit failure and inter-satellite link equipment failure, GEO satellite orbit determination accuracy can considerably decrease or orbit determination may fail, which undermines the positioning and timing service performance and availability of the BDS [18]. In case such anomalies occur during the orbit manoeuvre or recovery of GEO satellites, the kinematic orbit determination method is adopted for GEOs. L-band data are the only data resource, if the L-band data are abnormal in this period, the orbit error increases sharply and seriously affects RDSS service performance.

Currently, when the aforementioned problems occur, the out-station beam of this GEO satellite will be closed forcibly, and then the RDSS service of this GEO satellite will be stopped. However, only three GEO satellites of the BDS-3 provide the RDSS services, and the beam coverage areas of the three satellites are different. If the out-station beam of a GEO satellite is closed in the aforementioned situation, the RDSS services cannot be provided to users in the beam coverage area of that satellite. To solve the aforementioned problem, RDSS data can be used for GEO satellite orbit determination, which can be adopted as a backup method during GEO satellite orbit manoeuvre and recovery, improving the reliability of the GEO satellite orbit determination mode.

Dr. Xing adopted the RDSS data to make the orbit determination for the BDS-3 GEO satellite, and the orbit accuracy for GEO is better than 10 m [19]. In the BDS GEO satellite orbit determination method by RDSS data, the RDSS equipment time delay calibration is the first problem to be solved. Dr. Xing adopted the combined orbit determination method using the RNSS data and RDSS data to solve the RDSS equipment time delay. But before the time delay calibration, the time delay value of one RDSS station needs to be accurately calibrated off-line. Then, based on the RNSS data, the equipment time

delay value of the other RDSS stations can be solved simultaneously when the orbit and dynamical parameters are solved in the combined orbit determination. Since the RNSS equipment also has a time delay, it is necessary to adopt SLR data or the user equivalent range error (UERE) method to regularly calibrate the RNSS equipment time delay [20], so it needs two equipment delay calibrations to achieve the accurate RDSS equipment time delay calibration.

This study proposes a new precise orbit determination (POD) method for GEO satellites based on BeiDou short-message communication ranging data. The method uses high-precision SLR data as the benchmark for the calibration of RDSS equipment time delay. We need not calibrate the time delay value of a certain RDSS station before the time delay calibration. Based on the precisely calibrated RDSS equipment time delay, RDSS data are used only in the orbit determination process. Thus, the high dependency of the existing GEO satellite orbit determination method on RNSS data can be reduced, the continuity and reliability of the precise orbit products of BeiDou GEO satellites during the orbit manoeuvre and recovery will be improved, and the integrated application of short-message communication and navigation services of BDS will be realised.

In this paper, we first describe the new POD method for the GEO satellites based on RDSS data and the principle of solving time delay parameters of RDSS equipment based on the SLR data. Then, we present the experiments by the measured RNSS, RDSS, and SLR data of the BDS. Finally, we evaluate the accuracy of the estimated RDSS time delay value, the orbit accuracy of GEO satellites based on the observation residuals, overlap orbit differences, and orbit accuracy in radial direction based on the SLR data.

## 2. Measurement Model of RDSS

BDS adopts an active positioning mode in the RDSS system, which adopts 4-way ranging observations. There are C-band uplink signals and S-band downlink signals in the out-station beam. There are L-band uplink signals and C-band downlink signals in the in-station beam. The out-station C-band uplink signals are transmitted by the master control centre (MCC) to a GEO satellite, the S-band downlink signals are transformed by GEOs to the RDSS user, the L-band uplink signals are transformed by the RDSS user to all observable GEO satellites, and the C-band downlink signals are transformed by GEOs to the MCC. The MCC accomplishes the 4-way ranging observations and estimates the position of the RDSS user within 1 s [5,6]. Figure 1 shows the schematic of the RDSS.

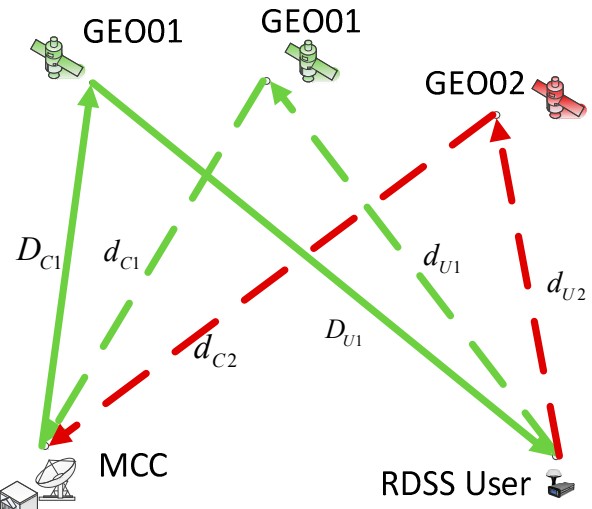

**Figure 1.** Schematic of the RDSS 4-way ranging observations.

The RDSS observation model is as follows:

$$\begin{aligned}
\rho_{S_1} &= D_{C1} + c \cdot \delta t_{C1O} + c \cdot \delta t_{S1O} + D_{U1} + c \cdot \delta t_U + d_{U1} + c \cdot \delta t_{S1I} + d_{C1} + c \cdot \delta t_{C1I} + \Delta E_{All1} + \varepsilon \\
\rho_{S_2} &= D_{C1} + c \cdot \delta t_{C1O} + c \cdot \delta t_{S1O} + D_{U1} + c \cdot \delta t_U + d_{U2} + c \cdot \delta t_{S2I} + d_{C2} + c \cdot \delta t_{C2I} + \Delta E_{All2} + \varepsilon \\
\tau_{all} &= \delta t_{C1O} + \delta t_{S1O} + \delta t_U + \delta t_{S1I} + \delta t_{C1I}
\end{aligned} \tag{1}$$

$$\begin{aligned}
D_{C1} &= \sqrt{(X^{S1}(t_1) - X_C)^2 + (Y^{S1}(t_1) - Y_C)^2 + (Z^{S1}(t_1) - Z_C)^2} \\
D_{U1} &= \sqrt{(X_U(t_2) - X^{S1}(t_1))^2 + (Y_U(t_2) - Y^{S1}(t_1))^2 + (Z_U(t_2) - Z^{S1}(t_1))^2} \\
d_{U1} &= \sqrt{(X^{S1}(t_3) - X_U(t_2))^2 + (Y^{S1}(t_3) - Y_U(t_2))^2 + (Z^{S1}(t_3) - Z_U(t_2))^2} \\
d_{C1} &= \sqrt{(X^{S1}(t_3) - X_C)^2 + (Y^{S1}(t_3) - Y_C)^2 + (Z^{S1}(t_3) - Z_C)^2} \\
d_{U2} &= \sqrt{(X^{S2}(t_4) - X_U(t_2))^2 + (Y^{S2}(t_4) - Y_U(t_2))^2 + (Z^{S2}(t_4) - Z_U(t_2))^2} \\
d_{C2} &= \sqrt{(X^{S2}(t_4) - X_C)^2 + (Y^{S2}(t_4) - Y_C)^2 + (Z^{S2}(t_4) - Z_C)^2}
\end{aligned} \tag{2}$$

where $\rho_{S_1}$, $\rho_{S_2}$ are the RDSS observation data for GEO01 and GEO02, respectively. $D_{C1}$ is the geometric range between MCC and GEO01 satellite in the out-station C-band uplink; $D_{U1}$ is the geometric range between GEO01 satellite and RDSS user in the out-station S-band downlink; $d_{U1}$ is the geometric range between GEO01 satellite and RDSS user in the in-station L-band uplink; $d_{C1}$ is the geometric range between MCC and GEO01 satellite in the in-station C-band downlink; $d_{U2}$ is the geometric range between GEO02 satellite and RDSS user in the in-station L-band uplink; $d_{C2}$ is the geometric range between MCC and GEO02 satellite in the in-station C-band downlink. $\varepsilon$ is the observation noise. $c$ is the speed of light. $\delta t_{C1O}$ is the transmission time delay in MCC in out-station downlink; $\delta t_{S1O}$ is the time delay of the out-station transferring link for GEOs; $\delta t_U$ is the transmission time delay of the RDSS user; $\delta t_{S1I}$, $\delta t_{S2I}$ are the time delay of the in-station transferring link for GEO01 and GEO02, respectively; $\delta t_{C1I}$ and $\delta t_{C2I}$ are the time delay of the receiver in MCC from GEO01 and GEO02, respectively. $\tau_{all}$ is the sum of the time delays of the out-station beam and the in-station beam, including the transmission time delay in MCC and the time delay of the out-station transferring link for GEOs, the transmission time delay of the RDSS user, the time delay of the in-station transferring link for GEOs and the time delay of the receiver in MCC. $\Delta E_{All1}$ is the sum of all errors, including the troposphere delays correction, the ionosphere delay correction, the Sagnac effect correction, the general relativistic correction in the out-station C-band uplink, the out-station S-band downlink, the in-station L-band uplink and the in-station C-band downlink and the antenna phase centre correction for GEOs, the RDSS user, and MCC receiver. $(X_C, Y_C, Z_C)$ and $(X_U, Y_U, Z_U)$ are the coordinates of RDSS MCC station and RDSS user, respectively, which are known quantities; $(X^{S1}, Y^{S1}, Z^{S1})$ and $(X^{S2}, Y^{S2}, Z^{S3})$ are the orbital positions of satellites GEO01 and GEO02, respectively.

## 3. Orbit Determination Method for GEO Satellite Based on RDSS Data

### 3.1. Principles of Orbit Determination

According to the RDSS measurement principle, the orbit accuracy of GEO satellites based on RDSS data considerably depends on the time delay accuracy of RDSS equipment. The time delay for the MCC and satellite is calibrated by a special instrument before being adopted in the BDS. The user equipment time delay is usually calibrated in the special equipment calibration field and the accuracy is better than 10 ns. Owing to equipment aging and hardware replacement, the time delay of RDSS equipment involves the problem of long-term drift [21–23]. The common method is recalibration and retesting; during this process, the equipment of the MCC needs to be off the line and the user equipment needs to be returned to the equipment calibration field for recalibration. Therefore, the time delay recalibration of RDSS equipment is tedious and time-consuming, potentially affecting the RDSS service of BDS, especially since the recalibration of the satellite equipment delay cannot become true [24–27]. For navigation satellites, with centimetre- or millimetre-scale precision, SLR is a high-accuracy measurement method that is independent of radio measurements

and is also an important means of evaluating orbit accuracy [10,28,29]. All the BeiDou satellites in orbit are equipped with laser reflectors, laying an important foundation for the accuracy evaluation and time delay calibration of the RDSS data in the BDS.

For the new method of POD for GEO satellites by only using the RDSS data, first the time delay of the RDSS equipment is calculated based on SLR data, and then in the POD process, the corrected RDSS time delay value is fixed. The satellite orbit and dynamic parameters are solved only by using the RDSS data.

### 3.2. Principles of Calibrating Time Delay for RDSS Equipments Based on SLR Data

For calibrating RDSS equipment time delay based on SLR data, first the SLR data and RDSS data are preprocessed, then a reasonable dynamical model is selected, and, finally, SLR and RDSS data are jointly used for orbit determination. The estimated parameters are the satellite orbit, dynamic parameters, and the RDSS equipment time delay. All GEO satellites are processed together. The normal equations of the SLR data and the RDSS data are formed, respectively. The former only contains the partial derivatives of the satellite orbital parameters and dynamic parameters, while the latter contains the partial derivatives of the satellite orbital parameters, the dynamic parameters, and the partial derivatives of the RDSS time delay parameters. The final normal equations are formed by superpositioning both normal equations, and the joint orbit determination is completed through the least square method. The process flow chart of calibrating RDSS equipment time delay based on SLR data is shown in Figure 2.

For SLR observation $r_m^i$ and RDSS four-way ranging observation $\rho_{CU}^{ij}$, with the error term removed, the observation equations are as follows:

$$
\begin{aligned}
r_m^i &= 2 * \left| \mathbf{X}_m(t) - \mathbf{X}^i(t) \right| \\
\rho_{CU}^{ij} &= \left| \mathbf{X}_C(t) - \mathbf{X}^i(t) \right| + \left| \mathbf{X}_U(t) - \mathbf{X}^i(t) \right| + \left| \mathbf{X}_U(t) - \mathbf{X}^j(t) \right| + \left| \mathbf{X}_U(t) - \mathbf{X}^j(t) \right| + c \cdot \tau_{all}
\end{aligned}
\tag{3}
$$

where $\mathbf{X}_m(t)$ is the coordinates of the SLR station $m$, $\mathbf{X}_C(t)$ and $\mathbf{X}_U(t)$ are the coordinates of RDSS MCC station $k$ and user $l$, respectively, which are known quantities; $\mathbf{X}^i(t)$ and $\mathbf{X}^j(t)$ are the orbital positions of satellites $i$ and $j$, respectively; $\tau_{all}$ is sum of the time delays of the out-station beam and the in-station beam.

The error equation can be concluded by linearising Formula (4).

$$
V(t) = \begin{bmatrix} \dfrac{\partial r_m^i}{\partial \mathbf{X}^i(t_0)} & 0 & 0 \\ \dfrac{\partial \rho_{CU}^{ij}}{\partial \mathbf{X}^i(t_0)} & \dfrac{\partial \rho_{CU}^{ij}}{\partial \mathbf{X}^j(t_0)} & 1 \end{bmatrix} \begin{bmatrix} \mathbf{X}^i(t_0) \\ \mathbf{X}^j(t_0) \\ c \cdot \tau_{all} \end{bmatrix} - L(t)
\tag{4}
$$

where $\mathbf{X}^i(t_0)$ and $\mathbf{X}^j(t_0)$ are the initial orbital positions, velocities, and dynamic model parameters of satellites $i$ and $j$, respectively. Based on Equation (3), the RDSS equipment time delay is estimated using the least-squares batch method.

Because SLR observation is subject to weather conditions, it is impossible to achieve all-weather observations. Therefore, the POD method by SLR and RDSS data cannot be a routine method for GEOs. While RDSS data can be obtained in real time, this paper adopts the SLR data to calibrate the time delay of the RDSS. RDSS data are used only for GEO satellite orbit determination, for which only the parts corresponding to RDSS data in Formulas (1) and (2) are used.

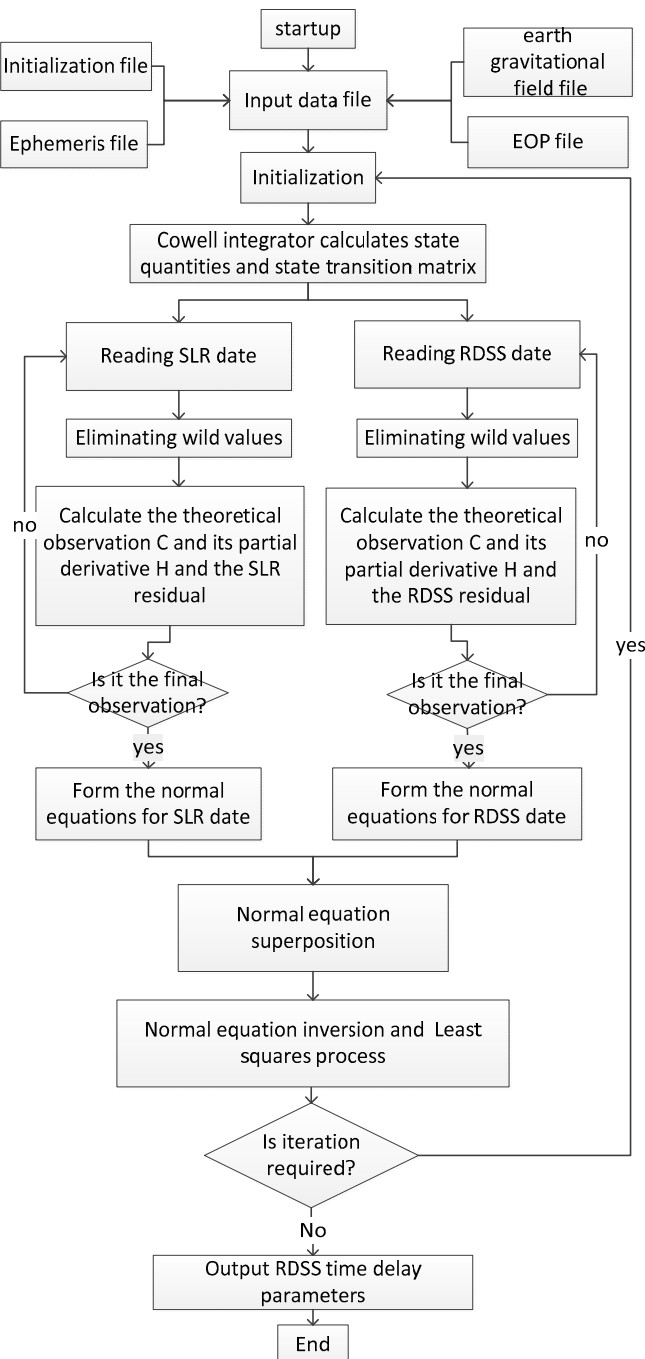

**Figure 2.** Flow chart of calibrating RDSS equipment time delay based on SLR data.

### 3.3. Program Design

The RDSS data are obtained from seven RDSS calibration stations deployed in Beijing, Sichuan, Hainan, Southeast China, and Northeast China, and two in Xinjiang. The SLR station was deployed in Beijing. The arc length for orbit determination is three days, and the data sampling rate is 60 s. The estimated parameters include the GEO initial orbit parameters of GEO satellites, solar radiation pressure parameters, and RDSS equipment time delay parameters. The RDSS equipment time delay is the sum of outstation and in-station link equipment time delays $\tau_{all}$. The normal equations of SLR data and the RDSS four-way ranging data are formed. Based on these, the pseudorange/pseudophase data and the RDSS four-way ranging data can be processed independently, thus flexibly processing different types of data.

As for the SLR data preprocessing, first, the phase centres of the antennas of tracking stations are corrected; second, the phase centres of the antennas that transmit satellite navigation signals are attributed to the centre of mass of the satellite; third, the Marini model is used to deduct the tropospheric error; and finally, the errors of general relativity and the tidal effect are corrected. In the RDSS data preprocessing, the phase centres of the satellite and station antenna are corrected; then, the Black model is used to correct the errors of tropospheric delay and the 14-parameter Klobuchar model is used to correct the ionospheric delay; finally, the error terms of the Earth rotation effect and general relativity are corrected.

The orbits for BDS-2 5 GEO satellites, BDS-3 3 GEO satellites are determined. Table 1 shows the strategies for the combined POD methods by RDSS and SLR data.

**Table 1.** Strategies for the combined POD methods by RDSS and SLR data.

| | POD Methods by RDSS and SLR Data |
|---|---|
| Satellite | 8 GEO: C01, C02, C03, C04, C05 in BDS-2 system C59, C60, C61 in BDS-3 system |
| Stations | Seven RDSS calibration stations deployed in Beijing, Sichuan, Hainan, Southeast China, and Northeast China, and two in Xinjiang One SLR station: Beijing |
| Arc length | 3 days, sampling interval: RDSS data: 60 s SLR data: 1 s |
| observation | RDSS data in Seven RDSS calibration stations SLR data in Beijing station |
| Estimated parameter | GEO Initial orbit of the satellite, solar radiation pressure parameters and RDSS equipment time delay parameters |
| Parameter estimation method | Least squares algorithm |
| Gravitational field model | EGM 2008 12 × 12 |
| Sun, Moon gravity and the gravitational force of the other planets | Jet Propulsion Laboratory Development Ephemeris 405 (JPL DE405) |
| Solar radiation pressure (SRP) | An empirical SRP model which is similar to BERNESE ECOM 9 parameter model |
| Solid tides, ocean tide perturbation | IERS Convention 2003 |
| Precession and nutation | IAU2000R06 |
| EOP parameters | Constraints to the International Earth Rotation and Reference Systems Service (IERS) C04 model |

## 4. Experiment and Analysis

The RDSS data are from 1 to 3 July 2019 and from 1 June to 31 July 2021. The SLR data are on 2 July 2019 and from 1 June to 31 July 2021 in Beijing. These data are jointly used for the time delay calibration of the RDSS equipment, and the RDSS data from 1 to 10 July 2019 and from 29 May to 31 July are used for sliding orbit determination with a 3-day arc length and orbit determination. We evaluate the accuracy of the estimated RDSS time delay value, the orbit accuracy of GEO satellites based on the observation residuals, overlap orbit differences, and orbit accuracy in radial direction based on the SLR data.

### 4.1. Accuracy Analysis of RDSS Time Delay

Assume that the out-station beam number of an RDSS four-way range is $i$, the in-station beam number is $j$, and the channel number of the in-station beam is k. Then, the time delay number of the RDSS data is $240 \times (i - 1) + 24 \times (j - 1) + k$. Since BDS-2 has 10 out-station beams, 10 in-station beams, and 24 channels in one in-station beam, the time

delay number ranges from 1 to 2400. For satellite C01, the out-station beam number *i* is 1 or 2, and the in-station beam number *j* is 1 or 2. If C01 is used as the out-station and in-station satellite simultaneously, the time delay value is within the following four ranges.

$$
num = \begin{cases} 0 \text{ to } 24 & i = 1, j = 1 \\ 24 \text{ to } 48 & i = 1, j = 2 \\ 240 \text{ to } 264 & i = 2, j = 1 \\ 264 \text{ to } 288 & i = 2, j = 2 \end{cases}
\tag{5}
$$

The out-station beam number *i* for satellite C02 is 3 or 4, and the in-station beam number *j* is 3 or 4. Therefore, the time delay number of satellites C01 to C05 can be obtained when they are both out-station and in-station satellites. For C01 to C05 satellites, the time delay number of the RDSS data is as follows:

$$
num = \begin{cases} 0 \text{ to } 48 & and & 240 \text{ to } 288 & for \text{ C01} \\ 528 \text{ to } 576 & and & 768 \text{ to } 816 & for \text{ C02} \\ 1056 \text{ to } 1104 & and & 1298 \text{ to } 1344 & for \text{ C03} \\ 1584 \text{ to } 1632 & and & 1824 \text{ to } 1872 & for \text{ C04} \\ 2112 \text{ to } 2160 & and & 2352 \text{ to } 2400 & for \text{ C05} \end{cases}
\tag{6}
$$

The SLR observations of satellites C01 to C05 were carried out in Beijing. The values and standard deviation (std) of the RDSS combined time delay solved in the POD based on SLR and RDSS data from 1 to 3 July are shown in Figure 3. In the figure, the horizontal axis represents the combined time delay number.

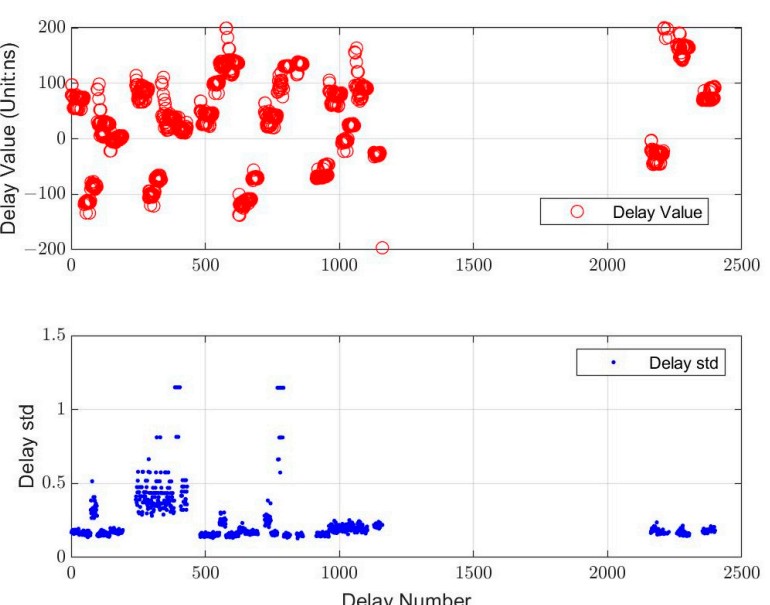

**Figure 3.** RDSS time delay value solution and post-testing standard deviation.

As the figure indicates, some of the combined time delay parameters cannot be estimated because there are no corresponding RDSS observation data. For example, as satellite C04 does not have in-station and out-station data, the combined time delay parameters of this satellite have no solution. The std of most of the combined delay is less than 1 ns, and the std of some time delay is greater than 1 ns but less than 1.5 ns because of insufficient observation data, which means the internal consistency of the time delay is better than 1 ns. The estimated combined time delay value is within ±200 ns.

### 4.2. Observation Residuals of RDSS Data

The POD experiments by RDSS data are only made by fixing the RDSS time delay value based on SLR data calibration. Only the satellite orbit and dynamic parameters are solved. The residual sequence of the RDSS data from 1 to 3 July is shown in Figure 4. The root mean square (RMS) value of the RDSS residuals for the GEO satellite is 1.74 m. The residual magnitude is decided by the observation accuracy of the data type. Because the observation accuracy of RDSS data is 1 to 2 m, the residual magnitude is normal.

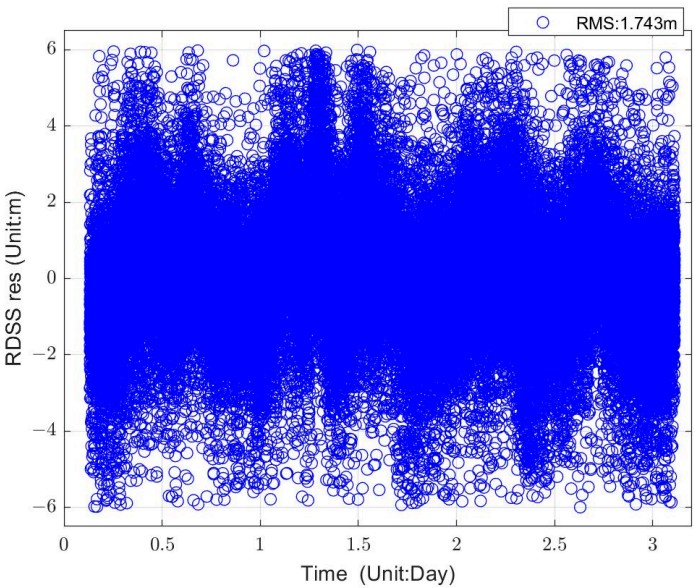

**Figure 4.** Observation Residuals of RDSS Data.

### 4.3. Overlap Orbit Differences (OOD)

The International GNSS Service (IGS) and the Multi-GNSS Experiment (MGEX) work together to provide the highest-quality GNSS data and products for free. Among the dozens of analysis centers, the BDS system precision products provided by Wuhan University (WHU, Wuhan, China) are the most complete. The SLR data internationally jointly measured were used to evaluate the orbit accuracy in radial direction, which was 40 cm for the BDS GEO satellites [29]. It is used as the reference orbit for evaluating the orbit accuracy in this paper. The differences between the orbits obtained from RDSS data-based orbit determination and the reference orbit at the same time and the same arc segment show the orbit accuracy by the new POD method based on RDSS data only, including in the orbital arc and the 24 h orbital prediction arc. The signal-in-space range error (*SISRE*) in orbit is determined by the orbit error in the *R* (radial), *T* (normal), and *N* (tangential) directions, which is as follows:

$$SISRE_{orbit} = \sqrt{1.0 \times \Delta R^2 + (0.09 \times \Delta T)^2 + (0.09 \times \Delta N)^2} \qquad (7)$$

The OOD Comparisons in the Orbital Arc

If orbit A is determined by the data on days n + 1, n + 2, and n + 3, orbit B is determined by the data on days n + 2, n + 3, and n + 4, and the orbital difference of the overlapping arc segments in the *R*, *T*, and *N* directions on days n + 2 and n + 3 are counted and used as the OOD comparison values for orbit A in the orbital arc. Figures 5 and 6 show the sketch map of the evaluation of the OOD comparisons in the orbital arc and the 2-h orbital prediction arc.

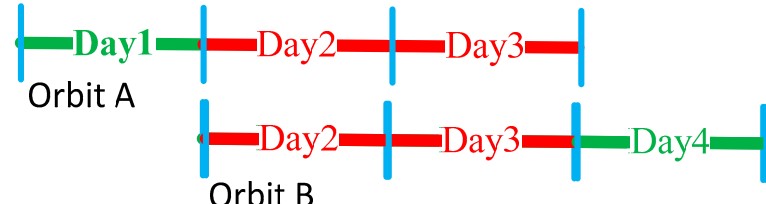

**Figure 5.** Evaluation of the OOD comparisons in the orbital arc. The red arc segments represent the overlapping arc segments.4.3.2. The OOD Comparisons in the 2-h Orbital Prediction Arc.

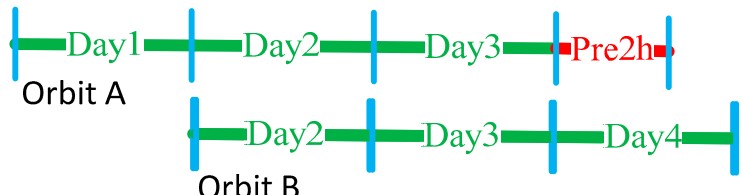

**Figure 6.** Evaluation of the OOD comparisons in the 2-h orbital prediction arc. The red arc segments represent the overlapping arc segments.

Orbit B is used as the reference orbit for evaluating the orbit accuracy in the 2-h orbital prediction arc of orbit A. The differences between the 2-h orbital prediction arc of orbit A and orbit B in the R, T, and N directions are counted and used as the OOD comparison value for orbit A in the 2-h orbital prediction arc.

Tables 2 and 3 give the OOD comparisons for C01, C02, C03, and C05 satellites in the orbital arc and the 2-h orbital prediction arc in the POD of 1 to 3 July 2021, respectively. Figures 7 and 8 give the OOD comparisons for the C02 satellite in the orbital arc and the 2-h orbital prediction arc, respectively. In these tables and figures, R, T, N, Positions and SISRE (orbit) represent the results in the radial direction, along-track direction, cross-track direction, the three-dimensional (3D) position, and SISRE in the orbital arc, respectively.

**Table 2.** Overlapping orbit differences for GEOs in the orbital arc (unit: m).

| Satellites | R | T | N | Positions | SISRE (Orbit) |
|---|---|---|---|---|---|
| C01 | 0.47 | 6.05 | 1.81 | 6.33 | 0.74 |
| C02 | 0.41 | 5.35 | 1.74 | 5.64 | 0.65 |
| C03 | 0.38 | 5.37 | 1.62 | 5.63 | 0.63 |
| C05 | 0.43 | 6.97 | 0.63 | 7.01 | 0.76 |
| C59 | 0.48 | 5.99 | 1.86 | 6.29 | 0.74 |
| C60 | 0.39 | 5.32 | 1.77 | 5.62 | 0.64 |
| C61 | 0.34 | 5.30 | 1.59 | 5.54 | 0.60 |
| Mean Values | 0.41 | 5.76 | 1.57 | 6.01 | 0.68 |

**Table 3.** Overlapping orbit differences for GEOs in the 2-h orbital prediction arc (unit: m).

| Satellites | R | T | N | Positions | SISRE (Orbit) |
|---|---|---|---|---|---|
| C01 | 0.56 | 6.73 | 2.25 | 7.12 | 0.85 |
| C02 | 0.53 | 6.49 | 2.78 | 7.09 | 0.82 |
| C03 | 0.40 | 5.55 | 2.54 | 6.12 | 0.68 |
| C05 | 0.58 | 7.91 | 2.56 | 8.34 | 0.94 |
| C59 | 0.57 | 6.77 | 2.28 | 7.17 | 0.86 |
| C60 | 0.51 | 6.46 | 2.74 | 7.03 | 0.81 |
| C61 | 0.36 | 5.47 | 2.5 | 6.03 | 0.65 |
| Mean Values | 0.50 | 6.48 | 2.52 | 6.99 | 0.80 |

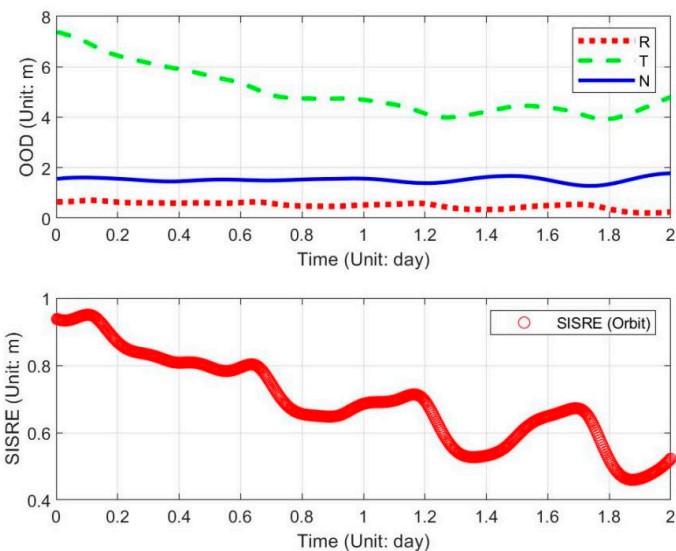

**Figure 7.** Overlapping orbit differences for C02 in the orbital arc.

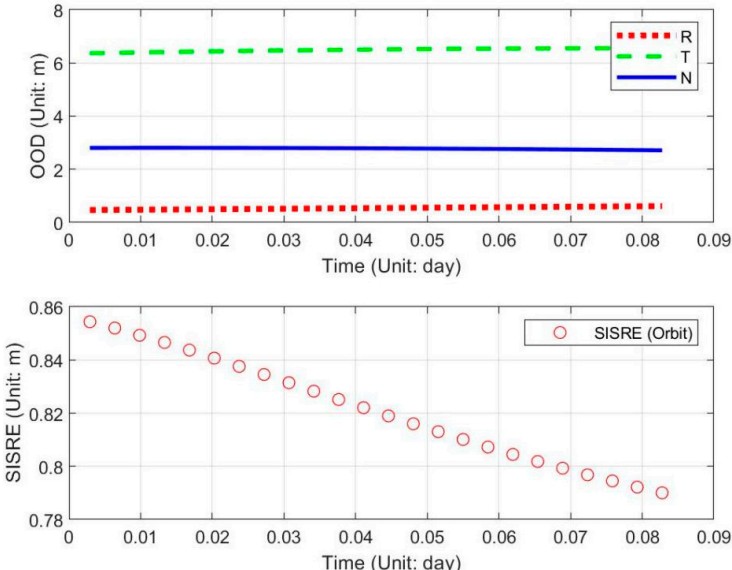

**Figure 8.** Overlapping orbit differences for C02 in the 2-h orbital prediction arc.

The tables and figures show that in the POD for GEO satellites based on only RDSS data, the position error and SISRE in orbit in the orbital arc for GEO satellites are 6.01 m and 0.68 m, respectively; the position error and SISRE in orbit in the 2-h orbital prediction arc are 6.99 m and 0.80 m, respectively. The orbit error in the radial direction is better than 0.6m, but those in the tangential and normal directions are about 6 m and 2 m, respectively. Due to the geostationary characteristics of GEO satellites, only domestic RDSS stations are used for orbit determination, so their geometric configuration is poor. The correlation between the RDSS measurement and the orbit in the radial direction is about 0.98. Those in the normal direction and the tangential direction are very small, especially that in the tangential direction, which is the smallest.

The 3D position error of GEO satellites using RNSS data in the orbital arc and the 2-h orbital prediction arc are 1.46 m and 2.03 m, respectively. The orbit results by the POD based on only RDSS data display slightly low precision due to noise from RDSS data and the ionospheric delay error after the correction by the 14-parameter Klobuchar model with a residual error of 1 to 2 m. The pseudorange data and phase data are used in the POD

by RNSS data. The measurement noise of the pseudorange data is about 30 cm and the measurement noise of the phase data is about 2 mm.

The orbit accuracy is in the order of C03, C02, C01, and C05 from high to low. This is related to the observation geometry. Since the domestic stations are approximately distributed from 75°E to 125°E, the C01, C02, C03, C04, and C05 satellites are, respectively, fixed at 140°E, 84°E, 110.5°E, 160°E, and 58.75°E, the station geometry of C03 satellite is the best. C59, C60 and C61 satellites in BDS-3 are fixed at 140°E, 84°E and 110°E, respectively, so the orbit accuracy is equivalent to that of the C01, C02, and C03 satellites.

The combined POD methods by RDSS and SLR data are adopted to solve the RDSS time delay in this paper. The stability of the time delay is important for the orbit accuracy for GEOs. Therefore, the POD experiments over 2 months from 1 June to 31 July 2021 are performed with the time delay estimated on 1 June 2021 to verify the applicability of the time delay and the orbit accuracy over multiple days. Figures 9 and 10 give the OOD results in the R, T, and N, SISRE for C03 and C60 satellites in the orbital arc and the 2-h orbital prediction arc in the POD experiments from 1 to 30 of June, respectively.

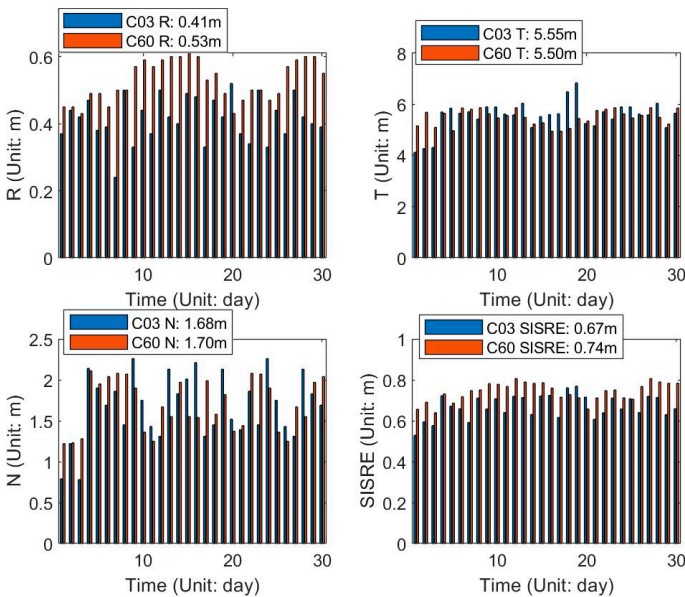

**Figure 9.** OOD results for C03 and C60 in the orbital arc in the 30 POD experiments.

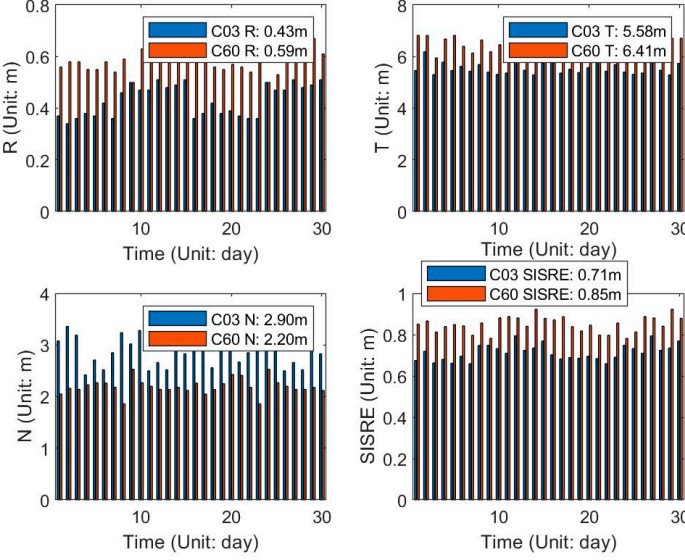

**Figure 10.** OOD results for C03 and C60 in the 2-h orbital prediction arc in the 30 POD experiments.

The figures show that in the 30 POD experiments, the mean value of the SISRE in orbit in the orbital arc for C03 and C60 satellites are 0.67 m and 0.74 m, respectively; which in the 2-h orbital prediction arc are 0.71 m and 0.85 m, respectively. This analysis shows that the multiday orbit accuracies with the adjusted time delay values are high and stable. The corrected RDSS time delay value is stable in about 30 days. This calibration method is applicable for at least 30 days. We can regularly calibrate the RNSS equipment time delay about every 1 or 2 months. When the orbit accuracy for GEO satellites in BDS is better than 10 m, the horizontal positioning accuracy at different locations averages about 9 m. Orbit accuracy using RDSS data must be suitable for a horizontal positioning precision of less than 20 m of BeiDou-3 RDSS service, with an orbit accuracy higher than 15 m. Therefore, the orbit accuracy achieved in this paper is far superior to the index requirement.

Next, the short-arc dynamic orbit determination experiments are made based on the 4-h RDSS data after the orbit manoeuvre to test the fast orbit recovery capability of GEO satellites after the orbit manoeuvre based on RDSS data. The precise ephemeris of MGEX is used as the benchmark. The experiments are made once in one day from 1 June to 30, 2021, the mean value of these 30 POD experiments is the last results. The orbit accuracies in the short-arc orbit determination for GEO satellites are shown in Table 4.

**Table 4.** Orbit accuracy in Short-arc orbit determination for GEO satellites based on RDSS data (unit: m).

| Satellites | R | T | N | Positions | SISRE (Orbit) |
|:---:|:---:|:---:|:---:|:---:|:---:|
| C01 | 1.95 | 8.76 | 5.33 | 10.44 | 2.16 |
| C02 | 2.21 | 9.78 | 6.05 | 11.71 | 2.44 |
| C03 | 2.14 | 9.32 | 6.14 | 11.29 | 2.36 |
| C04 | 2.80 | 10.14 | 7.60 | 12.98 | 3.02 |
| C05 | 2.78 | 10.06 | 7.55 | 12.88 | 3.00 |
| C59 | 1.90 | 8.65 | 5.13 | 10.23 | 2.10 |
| C60 | 1.78 | 8.14 | 4.89 | 9.66 | 1.97 |
| C61 | 1.67 | 8.13 | 5.03 | 9.71 | 1.88 |
| Mean values | 2.15 | 9.12 | 5.97 | 11.11 | 2.37 |

As the table shows, in the short-arc dynamic orbit determination based on RDSS data only for GEO satellites, the position accuracy for GEO satellites after 4 h of manoeuvring is 11.11 m. Although this is slightly inferior to the current RNSS data-based orbit determination accuracy during recovery, it can meet the aforesaid accuracy index requirement of 15 m.

### 4.4. Orbit Accuracy Analysis in Radial Direction Based on the SLR Data

The orbit accuracy in radial direction for satellites C01, C02, C03, and C05 was validated with SLR observation data, and the SLR residual details of satellites C01 and C03 are shown in Figures 11 and 12, respectively. The orbital accuracies in radial direction for satellites C01, C02, C03, and C05 based on RDSS data only are 0.547, 0.569, 0.510, and 0.617 m, respectively, with a mean value of 0.561 m. The accuracies are slightly inferior to the orbit accuracy of BeiDou GEO satellites based on the global network data. The large deviation of SLR residuals comes from the orbital error in the radial direction of GEO satellites. In this paper, the observation stations were deployed in regions in China, and there are only seven stations with a limited observation accuracy of RDSS data of 1 to 2 m. Due to the geostationary characteristics of GEO satellites, the constraints of RDSS data in the radial direction for GEO satellites are limited. Moreover, the constraints in the normal and tangential directions are very small by the satellite to ground measurement data for GEO satellites. Therefore, the orbit errors in the radial, normal, and tangential directions cause a large deviation in the SLR residual error in the radial direction.

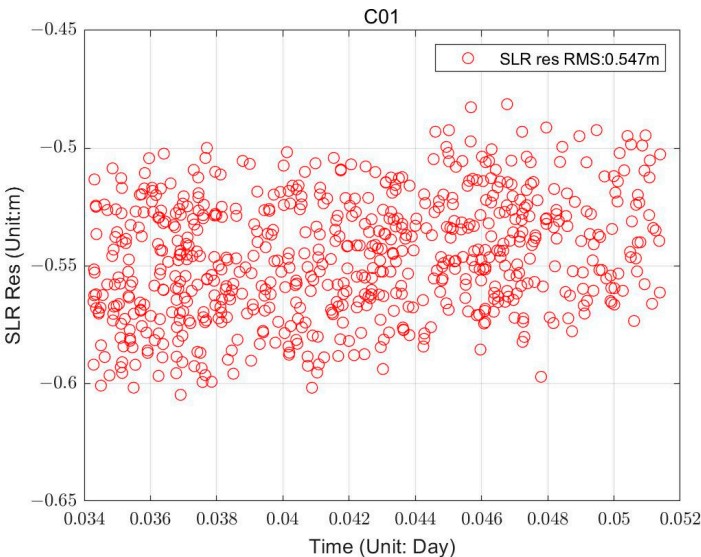

**Figure 11.** SLR residuals of the orbit for C01 satellite.

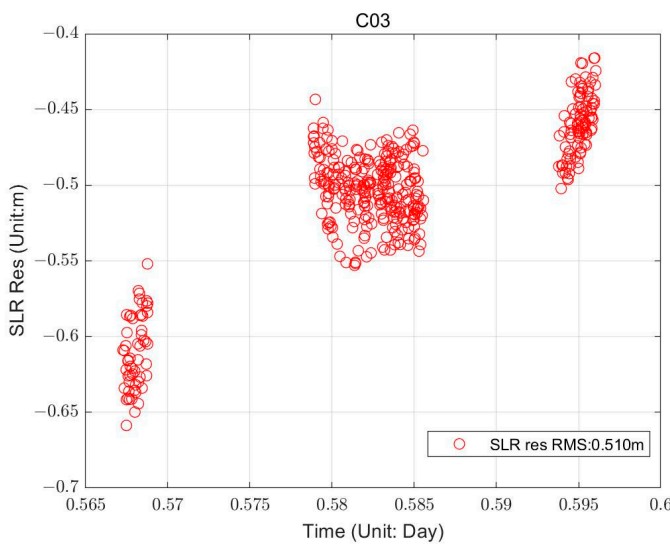

**Figure 12.** SLR residuals of the orbit for C03 satellite.

The SLR measurements for C01 to C05 satellites are carried out intermittently lasting 2 months from 1 June to 31 July 2021 in Beijing. The experiments are performed to verify the orbit accuracy analysis in a radial direction over multiple days. Table 5 gives the SLR residuals of the orbit for the C01–C05 satellites based on RDSS data. Figure 13 gives the SLR residuals for the C01 and C02 satellites.

**Table 5.** SLR residuals of the orbit for C01-C05 satellite based on RDSS data (unit: m).

| Satellites | 2 July 2019 | 1 June to 31 July 2021 |
|:----------:|:-----------:|:----------------------:|
| C01 | 0.55 | 0.56 |
| C02 | 0.57 | 0.54 |
| C03 | 0.51 | 0.50 |
| C04 | - | 0.55 |
| C05 | 0.62 | 0.57 |
| Mean values | 0.56 | 0.54 |

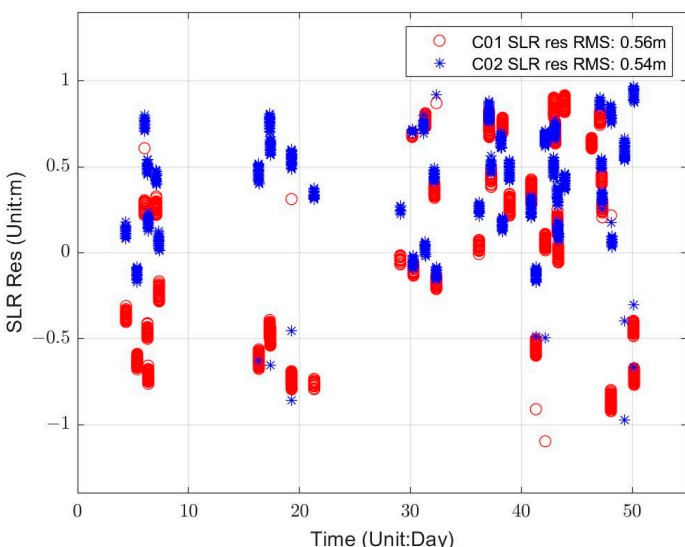

**Figure 13.** SLR residuals of the orbit for C01, C02 satellite from 1 June to 31 July 2021.

The tables and figures show that the SLR residuals are greater than −1 m but less than 1 m. The RMSs of the SLR residuals for C01 and C02 satellites are 0.56 m and 0.54 m, respectively. The RMS of the SLR residuals for GEO satellites in BDS-2 system is 0.54 m. By contrast, the accuracy of the RNSS phase data of the global network is about 40 cm. There are a large number of observation stations. With the development of the BDS, the improvement of observation accuracy of RDSS data, and the deployment of RDSS calibration stations on a wide scale, the orbit accuracy based on the RDSS data only for GEO satellites is expected to improve by a large margin.

## 5. Conclusions

In order to solve the problem of unstable orbital results of the BeiDou GEO satellites caused by RNSS data anomalies at different times, this study proposed a new POD for GEO satellites based on RDSS data only, and the time delay of RDSS equipment was calibrated by SLR data. The service performance and reliability of RDSS systems are improved. This study focused on the deep integration and application of short-message communication and navigation services of the BDS. Experiments based on SLR and RDSS data measured in the BDS were performed, and the main conclusions from this study are summarised as follows:

(1) The time delay accuracy of the RDSS data was better than 1 ns, while the observation accuracy of the RDSS was 1 to 2 m. Therefore, the time delay calibration accuracy meets the demand for time delay accuracy in the case of orbit determination for GEO satellites based on RDSS data.

(2) The OOD comparisons in the orbital arc and the 2-h orbital prediction arc for GEO satellites were 6.01 m and 6.99 m, respectively. The orbit accuracy for GEO satellites needs to be higher than 15 m in the RDSS service in BDS. Therefore, the orbit accuracy achieved in this study is considerably higher than this index.

(3) In the short-arc dynamic orbit determination based on RDSS data only for GEO satellites, the position accuracy after 4 h of manoeuvring is 11.11 m. The RDSS ranging data was used to achieve the orbit determination of GEO satellites when RNSS data were unavailable during the recovery period after manoeuvring. Thus, the POD methods of GEO satellites during the recovery period are enriched.

(4) The orbital accuracy in the radial direction based on the SLR data for GEO satellites was 0.54 m. The accuracy depends on the measurement accuracy of RDSS data and the current situation of regional station deployment. In the development of the BDS, we should strive for more and better satellites, orbits, and link resources. In addition, we should continuously improve the technical system of the RDSS and its measurement accuracy, as

well as widely deploy RDSS calibration stations to further improve the orbit accuracy of GEO satellites based on the RDSS data only.

(5) In view of the application of RDSS data in the orbit determination for BeiDou GEO satellites, this study has only conducted preliminary exploration. The orbit accuracy for GEO satellites based on RDSS is lower than that of GEO satellites based on RNSS data, which remains to be studied in depth in the next step. However, it should be noted that the new POD method based on RDSS data only for the GEO satellite proposed in this paper greatly expands the application scope of RDSS data. The functions of positioning, short-message communication, timing service, and orbit determination can be achieved based on RDSS data. The method also enriches the measurement approaches in the orbit determination for GEO satellites, which provides a backup orbit determination technology for BeiDou GEO satellites during the orbit manoeuvre or recovery period.

**Author Contributions:** Conceptualization, R.G. and J.C.; methodology, J.C. and R.G.; software, S.L.; validation, R.G. and J.C.; formal analysis, J.C; investigation, Z.C., J.X., J.G. and Y.T.; resources, R.G.; data curation, S.L.; writing—original draft preparation, X.L.; writing—review and editing, R.G. and J.C.; visualization, R.G., X.L. and J.C.; supervision, R.G. and J.C.; project administration, R.G.; funding acquisition, R.G. and X.L. All authors have read and agreed to the published version of the manuscript.

**Funding:** This research was funded by the National Natural Science Foundation of China (Grant Nos.: 41874043, 42004028, 41704037).

**Data Availability Statement:** Beijing Satellite Navigation Center provides all the test data used in this contribution, including the RDSS, SLR measurements, and the L-band measurements. All data will be made available for scientific research purposes by request to the Beijing Satellite Navigation Centre.

**Conflicts of Interest:** The authors declare no conflict of interest.

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
