# Peer review of "New Orbit Determination Method for GEO Satellites Based on BeiDou Short-Message Communication Ranging"

_remotesensing, doi:10.3390/rs14184602_

Round 1

Reviewer 1 Report

In this paper, the authors proposed a new orbit determination method for GEO satellites based on BeiDou short‐message communication ranging. It is a interesting topic and the study presents new discoveries. After providing more sufficient analysis and discussion, I suggest that the manuscript should be published.

1. I noticed that the RDSS data have been used in GEO satellite POD in previous studies, so what is the innovation of your new orbit determination method in the paper.

2. In Introduction,  please introduce the level of GEO satellite orbit determination based on L‐band data, which can be compared with the new orbit determination method in this paper.

3. Line 166, the corrected RDSS time delay value is constant or time-dependent value?

4. Figure 7 and Figure 8, how to explain the big deviations of SLR residuals.

5. Can you use more data to test and verify the method in this paper?

6. After Section 4, in my opinion, you should first have a discussion of your results, then look forward to it.

Reviewer 2 Report

Figure 3 represents observed delay with regard to delay number. In Eq. (5), you explain i is 1 or 2 for C01 satellite, i is 3 or 4 for C02 satellite, and so on. This means delay number is 0 to 239 for C01, 240 to 479 for C02, ..., doesn't it? You should clarify the relationship between delay number and satellite ID in Figure 3, for example, draw the range of delay number for each satellite.

Round 2

Reviewer 1 Report

This second revision meets my requirements.